# Measurement and spatiotemporal analysis of high-quality development of China's industry

**Yuexiang Yang[1], Lei Ren[1], Zhihui Du[2]\*, Guanqun Tong[3]**

1 School of Management, China University of Mining and Technology (Beijing), Beijing, China, 2 School of Banking and Finance, University of International Business and Economics, Beijing, China, 3 Canvard College, Beijing Technology and Business University, Beijing, China

\* 202000210048@uibe.edu.cn

**Data Availability Statement:** We uploaded the minimal data set as the supplementary material.

**Funding:** This work is supported by the Major Program of National Social Science Foundation of China No.18ZDA079. The funders had no role in

## Abstract

### Background

China's economy has been transitioning from a phase of rapid growth to high-quality development. The high-quality development of industry is the foundation of a sustainable and healthy growth of national economy, and is of great significance to improve people's living standards, and to meet people's needs for a better life.

### Methods

We develop an evaluation index system of high-quality development of industry from the perspectives of industrial benefit, innovation ability, coordination ability, green ability, opening ability and sharing ability. Based on a panel data of 30 provinces in China during 1999–2018, we evaluate the level of high-quality development of industry using the entropy-weight method and Technique for Order Preference by Similarity to an Ideal Solution (TOPSIS) method. Meanwhile we select six specific years and adopt the Natural Breaks method to classify the provinces according to their levels. At last, Moran's I index is used to analyze the spatial correlation among the provinces.

### Results

Opening ability and innovation ability are found to have greater impacts on industrial high-quality development than other indices, and their influence has been increasing in recent years. There are large spatial and temporal differences among different provinces. Municipalities and coastal provinces are found to be at constantly high levels. The levels in the central region dropped first and then increased, however it was the opposite in the western region. In the northeast region, the levels fluctuated greatly. Overall, the high-quality development of industry among China's provinces shows positive spatial correlation. Most provinces in China are in High-High and Low-Low clustering States. The High-High clustering type is mainly distributed in the eastern region and the Low-Low clustering type is mainly distributed in the western and central regions.

study design, data collection and analysis, decision to publish, or preparation of the manuscript.

**Competing interests:** The authors have declared that no competing interests exist.

## Conclusion

(1) Innovation ability and open ability are the most important factors. (2) Green ability has not sufficiently contributed to China's industrial development. (3) Regional and time evolution differences are significant. (4) There is a significant and stable spatial clustering effect in the high-quality development of industry among China's provinces.

## Introduction

In 2017, the 19th National Congress of the Communist Party of China proposed the concept of high-quality development for the first time, indicating that China's economy has been transitioning from a phase of rapid growth to high-quality development. "High-quality development is the development which can well meet people's growing needs for a better life. It is the development that embodies a new concept that innovation becomes the first driving force, coordination becomes the endogenous characteristic, green becomes the universal form, opening becomes the inevitable course, and sharing becomes the fundamental purpose." General secretary of the CPC Central Committee Xi Jinping said. From this, we can see that high-quality development means strong innovation ability, green environmental protection, harmonious social and economic relationship, and sharing of development achievements by the people, on top of the traditional concept of high input-output efficiency. General Secretary Xi Jinping also stressed that the requirements of high-quality development should be reflected in the economic, social, cultural, ecological and all other fields. The high-quality development of industry is the foundation of a sustainable and healthy growth of national economy, and is of great significance to improve people's living standards, and to meet people's needs for a better life.

The high-quality development of overall economy has been widely studied in the academia. Because of its wide range of contents, it can't be evaluated comprehensively using a single indicator or total factor productivity (TFP). Therefore, most studies establish indicator systems and use comprehensive evaluation methods. The concepts of innovative, coordinated, green, open and shared development (Five Concepts of Development hereafter) proposed by General Secretary Xi direct the construction of index systems at present, as in Li et al [1], Feng and Guo [2] and Liu and Guo [3]. Chen et al. [4] extend green development to sustainable development. Sun et al [5] think that the high-quality economic development can be measured from four aspects: economic vitality, innovation efficiency, green development and people's life. Wei and Li [6] construct an index system including economic structure optimization, innovation driven development, efficient resource allocation, perfect market mechanism, stable economic growth, regional coordination and sharing, excellent product and service, perfect infrastructure, construction of ecological civilization construction and economic achievements benefiting the mass of people.

Equal weight method [7–10] and entropy weight method [1, 4, 5, 11] are the most widely used methods of measurement. The equal weight method is used in the calculation of human development index and economic vulnerability index by the United Nations. This method is simple and transparent and can reflect the balanced influence of many factors on the high-quality economic development [10]. As an objective analysis method, Entropy weight method has broader applicability for analyzing and comparing the high-quality economic development in multiple time periods compared with other methods [11].

The high-quality development of industry is the pillar of the economy and has an important impact on the high-quality development of the whole national economy. Therefore, people pay more and more attention to the research of industrial high-quality development. There are mainly two measurement methods in the research of high-quality development of industry: one is to use Total Factor Productivity (TFP) or Green Total Factor Productivity (GTFP) as the measurement standard. The other is comprehensive evaluation.

The literature using total factor productivity as measurement standard include: Yang et al. improved the method of Fare et al. [12] by using the Global- Malmquist-Luenberger model and the fixed effect model [13]; Ju et al. [14] using the super efficiency DEA and Malmquist index model to produce the ETFP (Ecological Total Factor Productivity) as the industrial development level of major cities and urban clustering in the Yellow River Basin; and Zhao and Gu [15] integrating total factor productivity, GVC economic status index and export technology complexity, and comparing China with the United States. Since this method covers limited dimensions and is unable to measure the high-quality development of industry comprehensively, other studies construct index systems taking account of: the Five Concepts of Development [16–18]; economic performance, technological innovation, green development, quality, brand, integration of industrialization and informatization and high-end development as in Jiang et al [19]; efficiency, structure optimization, innovation drive and mode transformation as in Li and Wang [20]; input and output, environment, innovation, market factor and government factor as in Han and Ren [21]; talent accumulation as in Fu and Chu [22]; and energy security as in Hou et al. [23]. The entropy weight method is most widely used to evaluate the levels of indices [13, 18, 24, 25], along with analysis with coefficient of variation [19, 26], factor analysis [21], principal component analysis [27], projection pursuit model [20], etc.

In addition, some studies focus on the green dimension in the high-quality development of industry, such as Wang et al. [28] use the Entropy-weight TOPSIS model and the Theil index revealing the differences of industrial green development among three major urban clustering in China; Chen et al [29] develop an analytic hierarchy process to evaluate the levels of industrial green development of 30 provinces in China; Li et al. [30] use the panel data of 30 provinces in China from 2006 to 2015, analyzing the current situation, weaknesses and spatial-temporal differences of industrial green development in an improved evaluation framework.

According to the above review, we conclude four limitations in the measurement of high-quality development of industry. First, the Five Concepts of Development are the most consistent with China's high-quality development, which provide a reference for the construction of our comprehensive index system. The second is that the entropy weight and its improved methods are most widely used in the evaluation of levels of indices. Third, current research is lack of analysis from time and space dimensions, which cannot grasp the dynamic trend and spatial characteristics of high-quality development of industry in China. Fourth, current research is mainly based on a specific region or the whole country, and pays no attention to interprovincial analysis.

The Entropy weight TOPSIS method is characterized with simple calculation and a reasonable result and it combines the advantages of entropy weight and TOPSIS method, which makes the measurement results of high-quality economic development level more objective and reasonable [6]. Therefore, entropy weight TOPSIS method has been widely used in the field of high-quality development evaluation, Wei and Li [6], Du et al. [18], Ou et al. [31], Wang et al. [28] use it in the field of high-quality development evaluation. In addition, it is widely used in, sustainable development evaluation [32], efficiency evaluation [33, 34], contribution evaluation [35], people's life quality evaluation [36], competitiveness evaluation [37] and decision research [38].

This study focuses on and extends the Five Concepts for Development and introduces an industrial performance indicators system to build an evaluation index system for the high-quality development of industry in China. We measure the high-quality development of industry in 30 provinces in China from 1999 to 2018 by the entropy weight and TOPSIS method and use the spatial correlation analysis to reveal their spatial distribution and temporal evolution, and then we put forward countermeasures and suggestions. This paper originates in constructing a comprehensive index system and evaluating the high-quality development of industry from provincial and time dimensions, and reveal the interprovincial correlation among the provinces. We developed important references to promoting the high-quality development of industry in China.

The article is organized as follows. Section 1 reviews the current research on the measurement of high-quality development. Section 2 constructs the evaluation index system for high-quality development of industry. Section 3 introduces the calculation of entropy weight TOPSIS method and spatial correlation analysis. Empirical results are reported in Section 4, including the introduction of data sources and methods, the evaluation results of industrial high-quality development level, the sensitivity analysis of the measurement method and the results of spatial correlation analysis. Section 5 summarizes the conclusion, and puts forward policy suggestions.

## Construction of the evaluation index system

### Evaluation index system of high-quality development level of industry in China

The construction of our index system follows the principles of feasibility, independence, comparability and comprehensiveness in accordance with the Five Concepts of Development. On the basis of existing literature, we consult experts in the field through questionnaires and combine the Five Concepts of Development with traditional concept of industrial benefit. Our final index system covers the following six dimensions:

**Industrial benefit.** The essential objective of an enterprise is to make profits. Therefore, industrial benefit is the essential factor in measuring the high-quality development of industry. This dimension includes three secondary indices: profitability, operating ability and solvency.

**Innovation ability.** Continuous innovation of industrial technology is fundamental in upgrading the mode of development, improving economic efficiency, and realizing the transformation from "Made in China" to "Created in China". It is a top priority for the sustainable development of industry. Technological innovation is measured by input related indices including the effective input of R&D funds, talents and projects, as well as those related to output such as the effective output of patents of invention and new products.

**Coordination ability.** The coordination ability of industry development refers to the ability to balance development at macro and micro levels. At the macro level it refers to the coordination of industrial structure, measured by the proportion of high-tech industry and the coupling coordinative degree of high-tech and non-high-tech industry structure. At the micro level it refers to quality and production coordination. Quality coordination is measured as the ability to improve the quality of products and minimize cost loss due to quality problems, while production coordination is measured as the ability to improve production efficiency by improving total labor productivity.

**Green ability.** Green ability refers to the symbiotic ability of industrial development and environment. It is measured from two aspects: green production and green emission. Green

production is the ability to reduce energy consumption, wastewater and other residues in the production process. Green emission is the efficiency of industrial waste management.

**Opening ability.** Industrial opening refers to the introduction of foreign technology, capital and other resources and the strengthening of the output of industrial products. The introduction of foreign technology is measured by the proportion of foreign technology expenditure. The introduction of foreign capital resources is measured by the foreign capital opening index. And the output of industrial achievements is measured by the foreign trade opening index.

**Sharing ability.** Sharing ability refers to the ability to undertake a certain degree of social responsibility while obtaining economic benefits, so as to help solve the problems of social fairness and justice. There are wide-ranging ways to measure the sharing ability of enterprises. Due to data limit, this study evaluates social sharing, that is the tax liability born by the enterprises, and employee sharing, that is the ability to promote employment.

The index system is shown in Table 1.

## Pretreatment of indices

The concept of coupling coordinative degree is introduced as NO.11 in Table 1. It is calculated as follows [39]:

Suppose there are m provinces in the sample. $x_i$ and $y_i$ are the main business revenues of high-tech manufacturing enterprises above designated scale and non-high-tech manufacturing enterprises above designated scale in province $i$ respectively, where $I = 1, 2, \ldots m$.

①Data standardization

Scale high-tech manufacturing industrial enterprises: $f_i = \frac{x_i - \min\ x_i}{\max\ x_i - \min\ x_i}$,

Scale non-high-tech manufacturing industrial enterprises: $g_i = \frac{y_i - \min\ y_i}{\max\ y_i - \min\ y_i}$

②Calculation of coupling degree $C_i$:

Let $f_i$ and $g_i$ denote the main business revenues of scale high-tech and non-high-tech manufacturing enterprises in province $i$ respectively. The coupling degree between the two is as follows:

$$C_i = \sqrt{\frac{f_i \times g_i}{\left(\frac{f_i + g_i}{2}\right)}}$$

③Calculation of coupling coordinative degree $D_i$:

$D_i = \sqrt{C_i \times T_i}$, where $T_i = 0.5\,f_i + 0.5\,g_i$.

## Comprehensive evaluation method

Comprehensive evaluation methods include subjective weighting and objective weighting methods. The objective weighting methods determine the weights based on actual data and have the advantages of objectivity and transparency. They are applicable to our study since it is based on a large number of empirical data. As one of the most widely used objective weighting methods, entropy weight method determines the weight of all data information, which is more complete than the commonly used Principal Component Analysis [40]. Moreover, the entropy weight method does not require too much data, avoiding the shortcomings of the commonly used factor analysis which requires large quantities of sample data [41]. In addition, we

**Table 1. Evaluation index system of high-quality development level of industry in China.**

| Primary Indices | Secondary Indices | NO. | Tertiary Indices | Unit | Index Attribute |
|---|---|---|---|---|---|
| Industrial benefit | Profitability | 1 | cost-profit ratio = profit/cost and expense | % | + |
| | Solvency | 2 | asset-liability ratio = total liabilities/total assets | % | - |
| | Operating ability | 3 | current assets turnover = main business revenue/current assets | time(s)/year | + |
| Innovation ability | Innovation inputs | 4 | Proportion of R&D personnel = number of R&D staff/total of employees | % | + |
| | | 5 | Proportion of R&D expenses = R&D expenses/main business revenue | % | + |
| | | 6 | Proportion of enterprises with R&D department = number of enterprises with R&D department/total of enterprise | % | + |
| | Innovation achievements | 7 | Number of valid invention patents per capita = number of valid invention patents/number of R&D staff | patent(s)/person | + |
| | | 8 | Proportion of new product revenue = new product revenue/main business revenue | % | + |
| Coordination ability | Production coordination | 9 | Total labor productivity = industrial added value/number of employees | CNY/person | + |
| | Coordination of industrial structure | 10 | Proportion of high-tech industry = main business revenue of high-tech industry/industrial main business income | % | + |
| | | 11 | Synergy index of high-tech manufacturing industry = coupling coordinative degree of main business revenue between high-tech manufacturing industry and non-high-tech manufacturing industry | / | + |
| | Quality coordination | 12 | Quality loss rate = sum of internal loss and external loss cost of product quality/gross value of industrial output | % | - |
| | | 13 | Product high-grade rate = number of superior products of manufacturing enterprises/number of end products of manufacturing enterprises | % | + |
| Green ability | Green production | 14 | Energy consumption per unit of industrial added value = total industrial energy consumption/industrial added value | ton(s) of standard coal/Million CNY | - |
| | | 15 | Wastewater discharge per unit of industrial added value = industrial wastewater discharge/industrial added value | Ton(s)/Million CNY | - |
| | | 16 | Waste discharge per unit of industrial added value = production of general industrial solid waste/ industrial added value | Ton(s)/Million CNY | - |
| | Green emission | 17 | Utilization rate of solid waste treatment = comprehensive utilization of general industrial solid waste/production of general industrial solid waste | % | + |
| Opening ability | Trade openness | 18 | Trade openness = export delivery value/gross value of industrial output | % | + |
| | Openness of foreign capital | 19 | Openness of foreign capital = foreign paid-in capital of scale industrial enterprises/gross value of industrial output | % | + |
| | R&D openness | 20 | Proportion of expenditure on technology introduction = expenses on technology introduction/ total research cost | % | + |
| Sharing ability | Social sharing | 21 | Asset tax rate = total amount of enterprise tax/total assets | % | + |
| | Employee sharing | 22 | Employment ratio = employment in scale industrial enterprises/total employment | % | + |

introduce the TOPSIS to rank the high-quality development of industry of each province. Some researches show that the Entropy-weight TOPSIS method has better stability and robustness than the other methods. According to the analysis conducted by Zhang [42], when using the mean value of standard deviation as a measure of stability when the number of evaluation objects, indices or both change, Entropy-weight method is the best evaluation method for stability in the research area and TOPSIS method has good performance in the ranking method. Zhang [43] use Kendall harmony coefficient to analyze the robustness of different comprehensive evaluation methods when the dimensionless method and sample size changed. The results testify the high robustness of Entropy-weight method.

## Entropy-weight method

The calculation steps using the entropy-weight method are as follows:

Step 1. Establish the observation matrix.
 Set up m as the number of provinces and, establish an evaluation matrix including n indices. Let $x_{ij}$ denote the *j-th* index value of province i ($i = 1, 2, \ldots m; j = 1, 2, \ldots n$).

Step 2. Standardize the indices.
 In order to eliminate dimensions and realize comparability, the original data should be standardized. The method adopted is called "Range Standardization". Meanwhile we need to distinguish "Positive indicator" and "Negative indicator".
 Positive indicator:

$$y_{ij} = (x_{ij} - \min\ x_j)/(\max\ x_j - \min\ x_j),$$

Negative indicator:

$$y_{ij} = (\max\ x_j - x_{ij})/(\max\ x_j - \min\ x_j).$$

Step 3. Calculate the proportion of sample value in province *i* under the *j-th* indicator:

$$f_{ij} = \frac{y_{ij}}{\sum_{i=1}^{m} y_{ij}}.$$

Step 4. According to the definition of information entropy, calculate the entropy of the *j-th* index:

$$e_j = -k \sum_{i=1}^{m} f_{ij} \ln f_{ij},$$

Where $k > 0$, $k = 1/\ln m$, and denote if $f_{ij} = 0$, $f_{ij} \ln f_{ij} = 0$.
Step 5. Calculate the information entropy redundancy: $d_j = 1 - e_j, j = 1, 2, \ldots, m$.

Step 6. Calculate the weight of each index:

$$w_j = \frac{d_j}{\sum_{j=1}^{n} d_j}, j = 1, 2, \ldots, m.$$

## TOPSIS method

TOPSIS is a ranking method based on the proximity between a limited number of evaluation objects and idealized targets [44]. It evaluates the relative advantages and disadvantages among the existing objects. It is characterized by clear calculation process and operability. Ideal and negative ideal solutions are two important concepts in the method. Ideal solution refers to the most satisfactory solution for each attribute value, while negative ideal solution refers to the least satisfactory solution for each attribute value. The calculation is carried out as follows:

Step 1. Use the vector normalization method to calculate the normalized decision matrix.
 Suppose that the decision matrix of a multi-attribute decision making problem is $X = (x_{ij})_{m \times n}$, with *m* evaluation objects and *n* evaluation indices. The normalized decision

matrix is $Y = (y_{ij})_{m \times n}$, where

$$y_{ij} = \frac{x_{ij}}{\sqrt{\sum_{i=1}^{m} x_{ij}^2}}, \quad (i = 1, 2, \ldots, m; j = 1, 2, \ldots, n).$$

Step 2. Construct the weighted normal matrix $Z = (z_{ij})_{m \times n}$. According to the weight vector of each attribute given by policy makers, $w = [w_1, w_2, \ldots, w_n]^\tau$, then

$$z_{ij} = w_i \cdot y_{ij}, (i = 1, 2, \ldots, m; j = 1, 2, \ldots, n).$$

Step 3. Determine the best alternative $z_j^+$ and the worst alternative $z_j^-$:

The best alternative is:

$$z_j^+ \begin{cases} \max z_{ij} & (j \text{ denotes benefit attribute}) \\ \min z_{ij} & (j \text{ denotes cost attribute}) \end{cases}, \quad (j = 1, 2, \ldots, n).$$

The worst alternative is:

$$z_j^- \begin{cases} \max z_{ij} & (j \text{ denotes cost attribute}) \\ \min z_{ij} & (j \text{ denotes benefit attribute}) \end{cases}, \quad (j = 1, 2, \ldots, n).$$

Step 4. Calculate the distances from the target alternative to the best and worst conditions.

The distance from the target alternative to the best conditions is

$$d_i^+ = \sqrt{\sum_{j=1}^{n} (d_{ij} - d_j^+)^2}, \quad (i = 1, 2, \ldots n).$$

The distance from the target alternative to the worst conditions is

$$d_i^- = \sqrt{\sum_{j=1}^{n} (d_{ij} - d_j^-)^2}, \quad (i = 1, 2, \ldots n).$$

Step 5. Calculate the relative nearness degree (ND hereafter) between each scheme and the best alternative: $C_i = d_i^- / (d_i^- + d_i^+), \quad (i = 1, 2, \ldots m).$

Step 6. Rank the alternatives according to $C_i$. The larger value of $C_i$, the nearer the optimal level of distance of province $i$.

## Spatial correlation analysis

We introduce Moran's I index to study the spatial correlation of industrial high-quality development. Moran's I index can be divided into global Moran's I index and local Moran's I index.

The global Moran's I index reflects whether there is a spatial autocorrelation in the high-quality industrial development level of all provincial administrative regions, it is calculated as follows:

$$Moran's \ I = \frac{N}{S_0} \frac{\sum_{i=1}^{N} \sum_{j=1}^{N} \omega_{ij}(y_i - \bar{y})(y_j - \bar{y})}{\sum_{i=1}^{N} (y_j - \bar{y})^2}$$

Where $N$ denotes the number of space elements, $y_i$ and $y_j$ represent the observed value of variable $y$ in space units $i$ and $j$, $\bar{y}$ denotes the mean value of the variable $y$. $\omega_{ij}$ represents the elements in the spatial weight matrix. In this study, we use the spatial weight matrix based on *Rook* proximity; $S_0$ is the sum of all elements of the space weight matrix $S_0 = \sum_{i=1}^{n} \sum_{j=1}^{n} \omega_{ij}$. Moran's I is in the range of $[-1, 1]$. The positive value means positive spatial autocorrelation, negative value means negative spatial autocorrelation, and 0 means random distribution in space. The larger the absolute value is, the greater the spatial correlation is.

Local Moran's I index reflects whether there is a spatial autocorrelation between a certain province and other province. The local Moran's I index is calculated as follows:

$$Local\ Moran's\ I = \frac{y_i - \bar{y}}{S_i^2} \sum_{j=1,j\neq j}^{N} \omega_{ij}(y_i - \bar{y}),$$

where $S_i^2 = \frac{\sum_{j=1,j\neq i}^{N} \omega_{ij}}{N-1} - \bar{y}^2$, Other variables are defined as the same as in the above.

The local Moran's I index can be indicated by Moran's scatter plot. The Moran's scatter plot divides the observed values into four categories: the first, second, third and fourth quadrant corresponding to High-High, Low-High, Low-Low and High-Low respectively.

## Empirical study on the industrial high-quality development in China

### Data sources and processing

The data in this study cover 30 provinces in China during 1999–2018 (excluding the data of Tibet Autonomous Region, Hong Kong, Macao and Taiwan). We source data from China Statistical Yearbook, China Industrial Statistical Yearbook, China Science and Technology Statistical Yearbook, China Energy Statistical Yearbook and Statistics Bureau data of China's provinces and cities. Since the latest China Science and Technology Statistics Yearbook and China Industrial Statistics Yearbook were published in 2019 at the time of this study, the latest data that could be collected were from 2018. In addition, due to the change of statistical caliber in the statistical yearbooks, statistical errors are inevitable. Interpolation method or linear trend method is used in data processing, and near historical values are used to replace the result of linear trend method when it is significantly different from the actual situation.

Firstly, we use the entropy-weight method to measure the weights of 22 tertiary indices from 1999 to 2018. Then the average weight of each index is further calculated, as shown in Fig 1.

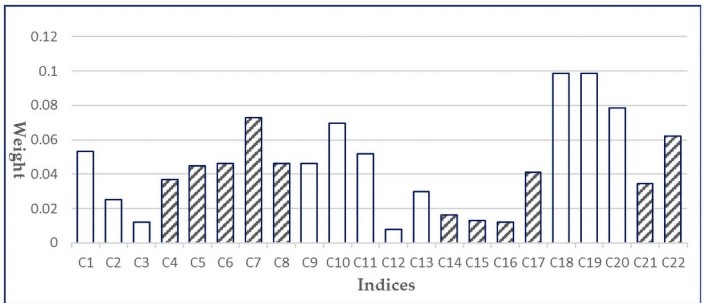

**Fig 1. The average weights of tertiary indices in China's industrial high quality development index system.**

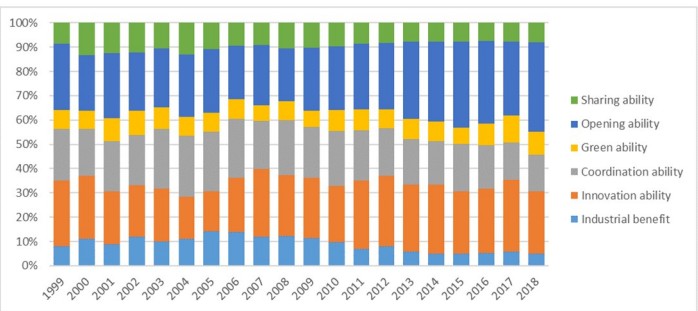

**Fig 2. Weights of primary indices in China's industrial high quality development index system.**

According to the principles of the entropy-weight method, larger index weights reveal more information and are more effective.

It can be seen that the indices such as number of valid invention patents per capita (C7), proportion of high-tech industries (C10), trade openness (C18), openness of foreign capital (C19), proportion of expenditure on technology introduction (C20) and employment ratio (C22) are weighted relatively heavy as they are all above 0.06. Among the indices of industrial benefit, cost-profit ratio (C1) is weighted significantly higher than that of the others. Among the indices of innovation ability, the weight of number of valid invention patents per capita (C7) is higher, while the weights of the other indices are similar to each other. For the coordination ability, the proportion of high-tech industry (C10) and high-tech manufacturing industry synergy index (C11) have the greatest weights. For green ability, the utilization rate of solid waste treatment (C17) weights the highest. For opening ability, the weights of trade openness (C18) and openness of foreign capital (C19) are relatively high. And for sharing capacity, the weight of employment ratio (C22) is relatively high.

The weights of primary indices are obtained by summing up the weights of their tertiary indices. The results are shown in Fig 2.

As can be seen from Fig 2, the weights of industrial benefit, coordination ability and sharing ability gradually increase and then decrease. Their final weights are less than their initial weights. The weights of opening ability decrease and then increase. Its final weight is significantly greater than its initial weight. The weights of innovation ability decrease firstly but then gradually stabilize and finally become consistent with the initial value. The change of weights of green ability over time is insignificant. When they are compared cross-sectionally, in the early and middle periods of time (1999–2012), the weights of innovation ability and opening ability are the greatest, followed by coordination ability. In the final period of time (2013–2018) the weights of opening ability are the greatest, followed by innovation ability and coordination ability.

In order to explore the dimensions that play a greater role in the current evaluation of high-quality develop3ment of industry, we calculate the weight of each primary index in 2018 respectively. The weight of primary index in 2018 is shown in Table 2.

**Table 2. Average weights of primary indices of industrial high-quality development.**

| Indices | Industrial benefit | Innovation ability | Coordination ability | Green ability | Opening ability | Sharing ability |
|---|---|---|---|---|---|---|
| Weight | 0.0506 | 0.2563 | 0.1514 | 0.0948 | 0.3681 | 0.0787 |

It can be seen from Table 2 that the ranking of the average weights of six primary indices is as follows: opening ability ≻ innovation ability ≻ coordination ability ≻ green ability ≻ sharing ability ≻ industrial benefit. Therefore, opening ability plays the most important role in the evaluation system of China's industrial high-quality development, followed by innovation ability and coordination ability.

## Analysis on the high-quality development level of China's industry

Based on the weights determined by the entropy-weight method, we use TOPSIS to calculate their relative nearness degrees and rank the high-quality development levels of 30 provinces during 1999–2018. Because there are too many years involved, we select six crucial years to analyze in detail, based on the characteristics of data and the time nodes for the five-year plans of national economic and social development. The selected years are 1999, 2001, 2006, 2011, 2016 and 2018. The years of 1999 and 2018 are the starting and end years of data respectively. The years of 2001, 2006, 2011 and 2016 are the starting years of the tenth, eleventh, twelfth and thirteenth five-year plans respectively. Their relative nearness degrees and rankings are shown in Table 3. Comprehensive results covering each year from 1999 to 2018 are attached in S1 Table.

First of all, based on the data of nearness degrees in Table 3 and using the natural breakpoint method, the levels of high-quality development of industry are divided into five categories, namely high, medium high, medium, medium low and low. The natural breakpoint method is one of data clustering methods, which realizes the best grouping of different categories by minimizing the average deviation within categories and maximizing the average deviation between categories. Results of the distribution of provinces in each high-quality development level category and their relative nearness degree ranges are shown in Table 4.

Five provincial regions have been at the high and medium high levels, namely Beijing, Shanghai, Jiangsu, Zhejiang and Guangdong. Tianjin, Fujian and Hainan have been at the high, medium high and medium levels. The provinces that have been at the medium high, medium and medium low levels include Shandong and Hubei. Shanxi, Inner Mongolia, Liaoning, Jilin, Jiangxi, Henan, Hunan, Sichuan, Yunnan, Qinghai and Xinjiang were always at the levels of medium, medium low and low. The regions that have been at the medium low and low levels include Hebei, Guangxi, Guizhou, Shaanxi, Gansu and Ningxia. In addition, Heilongjiang, Anhui and Chongqing have experienced relatively large fluctuations that involved four levels.

We analyze the level of high-quality development of the industry among the 30 provinces. Shandong, Jiangsu, Shanghai, Zhejiang, Fujian and Guangdong belong to the southeast coastal area, Shanxi, Anhui, Jiangxi, Henan, Hubei and Hunan belong to the central area, and Guangxi, Chongqing, Sichuan, Guizhou Yunnan, Shaanxi, Gansu, Qinghai, Ningxia and Xinjiang belong to the western area, Beijing, Tianjin, Hebei, Shanxi and Inner Mongolia belong to the north area, and Liaoning, Jilin and Heilongjiang belong to the northeast area. It can be found that:

Firstly, the distributions are not balanced. The provinces with higher levels of development are mainly municipalities directly under the central government, or the ones in the southeast coastal areas. The overall pattern is "high in the East, medium in the Middle and Northeast, and low in the Northwest".

Secondly, there are great differences in the evolution of high-quality industrial development levels among different regions in China. The development levels in the southeast coastal areas were constantly high, while those in the other areas experienced more tortuous evolution processes.

**Table 3. Relative nearness degrees and rankings of 30 provinces in 1999, 2001, 2006, 2011, 2014 and 2018.**

| | 1999 | | 2001 | | 2006 | | 2011 | | 2016 | | 2018 | |
|---|---|---|---|---|---|---|---|---|---|---|---|---|
| Province | ND | Rank | ND | Rank | ND | Rank | ND | Rank | ND | Rank | ND | Rank |
| Beijing | 0.4799 | 3 | 0.4095 | 7 | 0.3634 | 6 | 0.5719 | 4 | 0.4704 | 2 | 0.2843 | 3 |
| Tianjin | 0.3815 | 7 | 0.4955 | 5 | 0.3804 | 4 | 0.5359 | 5 | 0.2929 | 8 | 0.2174 | 8 |
| Hebei | 0.1864 | 22 | 0.1919 | 24 | 0.1436 | 24 | 0.1812 | 24 | 0.1236 | 24 | 0.0993 | 22 |
| Shanxi | 0.1170 | 27 | 0.1757 | 25 | 0.1514 | 22 | 0.1679 | 27 | 0.1749 | 17 | 0.1160 | 18 |
| Inner Mongolia | 0.1776 | 23 | 0.2700 | 13 | 0.1415 | 26 | 0.1659 | 28 | 0.1824 | 14 | 0.1981 | 9 |
| Liaoning | 0.2811 | 15 | 0.2783 | 12 | 0.2010 | 14 | 0.2653 | 11 | 0.2351 | 10 | 0.1854 | 10 |
| Jilin | 0.1997 | 21 | 0.2013 | 23 | 0.1559 | 21 | 0.1939 | 22 | 0.1935 | 11 | 0.0912 | 25 |
| Heilongjiang | 0.2871 | 14 | 0.3590 | 8 | 0.2777 | 10 | 0.2206 | 18 | 0.1229 | 25 | 0.0948 | 24 |
| Shanghai | 0.4121 | 6 | 0.5374 | 2 | 0.4198 | 3 | 0.7780 | 1 | 0.7630 | 1 | 0.8018 | 1 |
| Jiangsu | 0.3205 | 10 | 0.4264 | 6 | 0.3734 | 5 | 0.5972 | 3 | 0.3707 | 6 | 0.2586 | 5 |
| Zhejiang | 0.4233 | 4 | 0.5152 | 3 | 0.3303 | 8 | 0.4666 | 7 | 0.3108 | 7 | 0.2759 | 4 |
| Anhui | 0.3151 | 11 | 0.2302 | 16 | 0.1476 | 23 | 0.2296 | 15 | 0.1891 | 12 | 0.1634 | 12 |
| Fujian | 0.5414 | 1 | 0.5001 | 4 | 0.3474 | 7 | 0.4432 | 8 | 0.2373 | 9 | 0.1777 | 11 |
| Jiangxi | 0.2512 | 17 | 0.1643 | 28 | 0.1739 | 16 | 0.2345 | 14 | 0.1666 | 18 | 0.1563 | 13 |
| Shandong | 0.3303 | 9 | 0.3164 | 9 | 0.2393 | 12 | 0.2816 | 10 | 0.1841 | 13 | 0.1344 | 15 |
| Henan | 0.2675 | 16 | 0.1524 | 30 | 0.1331 | 27 | 0.1805 | 26 | 0.1408 | 21 | 0.1120 | 19 |
| Hubei | 0.3107 | 12 | 0.2187 | 20 | 0.1701 | 18 | 0.2502 | 12 | 0.1780 | 15 | 0.1289 | 16 |
| Hunan | 0.2920 | 13 | 0.2281 | 18 | 0.1628 | 20 | 0.2255 | 16 | 0.1753 | 16 | 0.1435 | 14 |
| Guangdong | 0.5072 | 2 | 0.5665 | 1 | 0.4232 | 2 | 0.6105 | 2 | 0.4242 | 4 | 0.3593 | 2 |
| Guangxi | 0.2175 | 20 | 0.2094 | 22 | 0.1329 | 28 | 0.1809 | 25 | 0.1273 | 23 | 0.0903 | 26 |
| Hainan | 0.3385 | 8 | 0.2927 | 11 | 0.5813 | 1 | 0.5094 | 6 | 0.3913 | 5 | 0.2359 | 6 |
| Chongqing | 0.4233 | 5 | 0.2295 | 17 | 0.1659 | 19 | 0.3129 | 9 | 0.4487 | 3 | 0.2338 | 7 |
| Sichuan | 0.1714 | 24 | 0.2121 | 21 | 0.1715 | 17 | 0.2501 | 13 | 0.1489 | 20 | 0.1162 | 17 |
| Guizhou | 0.1157 | 28 | 0.1646 | 27 | 0.1429 | 25 | 0.1891 | 23 | 0.1136 | 26 | 0.0889 | 27 |
| Yunnan | 0.2478 | 18 | 0.3026 | 10 | 0.2449 | 11 | 0.2167 | 19 | 0.1326 | 22 | 0.1114 | 20 |
| Shaanxi | 0.1687 | 25 | 0.2376 | 15 | 0.1777 | 15 | 0.2097 | 20 | 0.1548 | 19 | 0.1007 | 21 |
| Gansu | 0.0865 | 30 | 0.1525 | 29 | 0.1010 | 30 | 0.2255 | 17 | 0.0864 | 30 | 0.0678 | 30 |
| Qinghai | 0.1147 | 29 | 0.2224 | 19 | 0.2341 | 13 | 0.1474 | 30 | 0.0942 | 29 | 0.0814 | 29 |
| Ningxia | 0.1566 | 26 | 0.1688 | 26 | 0.1223 | 29 | 0.1959 | 21 | 0.1031 | 27 | 0.0850 | 28 |
| Xinjiang | 0.2192 | 19 | 0.2660 | 14 | 0.2795 | 9 | 0.1524 | 29 | 0.1024 | 28 | 0.0958 | 23 |

The high-quality development levels in the central region decreased from 1999 to 2001, and then increased especially in the periods of the eleventh and twelfth five-year plans (2006–2015). This improvement is closely related to the supporting strategies proposed by the central government in 2006 and continuously promoted by the State Council.

Except Chongqing, the levels in provinces in the western region have been rising and then declining. The rising period is from 1999 to 2005, in which 2001–2005 is the tenth five-year plan. The declining period is during the eleventh and twelfth five-year plans (2006–2015). The main reason for this change is that the western region development strategy was formally put forward in 1999, which helped it develop rapidly, but it did not catch up with more developed areas in the later period. However Chongqing is an exemption. We found that the levels in Chongqing have been high despite of some fluctuations. This could be explained by that it is a municipality directly under the central government and so entitled to more policy benefits.

The levels in North China varied significantly across regions. In Beijing and Tianjin they were constantly high, while in Hebei and Shanxi they were at medium and medium low

**Table 4. Distribution of provinces in each high-quality development level category.**

| Year | High level | Medium high level | Medium level | Medium low level | Low level |
|------|-----------|-------------------|--------------|------------------|-----------|
| 1999 | (0.3815–0.5414]:Fujian, Guangdong, Beijing, Zhejiang, Chongqing, Shanghai (6) | (0.2920–0.3815]:Tianjin, Hainan, Shandong, Jiangsu, Anhui, Hubei (6) | (0.2192–0.2920]:Hunan, Heilongjiang, Liaoning, Henan, Jiangxi, Yunnan (6) | (0.1170–0.2192]:Xinjiang, Guangxi, Jilin, Hebei, Inner Mongolia, Sichuan, Shaanxi, Ningxia (8) | [0.0865–0.1170]:Shanxi, Guizhou, Qinghai, Gansu (4) |
| 2001 | (0.4264–0.5666]: Guangdong, Shanghai, Zhejiang, Fujian, Tianjin (5) | (0.3164–0.4264]: Jiangsu, Beijing, Heilongjiang (3) | (0.2376–0.3164]: Shandong, Yunnan, Hainan, Liaoning, Inner Mongolia, Xinjiang (6) | (0.1919–0.2376]:Shaanxi, Anhui, Chongqing, Hunan, Qinghai, Hubei, Sichuan, Guangxi, Jilin (9) | [0.1524–0.1919]:Hebei, Shanxi, Ningxia, Guizhou, Jiangxi, Gansu, Henan (7) |
| 2006 | (0.4232–0.5813]:Hainan (1) | (0.2795–0.4232]: Guangdong, Shanghai, Tianjin, Jiangsu, Beijing, Fujian, Zhejiang (7) | (0.2010–0.2795]: Xinjiang, Heilongjiang, Yunnan, Shandong, Qinghai (5) | (0.1514–0.2010]: Liaoning, Shaanxi, Jiangxi, Sichuan, Hubei, Chongqing, Hunan, Jilin (8) | [0.1010–0.1514]:Shanxi, Anhui, Hebei, Guizhou, Inner Mongolia, Henan, Guangxi, Ningxia, Gansu (9) |
| 2011 | (0.5719–0.7780]:Shanghai, Guangdong, Jiangsu (3) | (0.3129–0.5719]:Beijing, Tianjin, Hainan, Zhejiang, Fujian (5) | (0.2345–0.3129]: Chongqing, Shandong, Liaoning, Hubei, Sichuan (5) | (0.1959–0.2345]: Jiangxi, Anhui, Hunan, Gansu, Heilongjiang, Yunnan, Shaanxi (7) | [0.1474–0.1959]:Ningxia, Jilin, Guizhou, Hebei, Guangxi, Henan, Shanxi, Inner Mongolia, Xinjiang, Qinghai (10) |
| 2016 | (0.4704–0.7630]:Shanghai (1) | (0.2929–0.4704]:Beijing, Chongqing, Guangdong, Hainan, Jiangsu, Zhejiang (6) | (0.1666–0.2929]:Tianjin, Fujian, Liaoning, Jilin, Anhui, Shandong, Inner Mongolia, Hubei, Hunan, Shanxi (10) | (0.1136–0.1666]:Jiangxi, Shaanxi, Sichuan, Henan, Yunnan, Guangxi, Hebei, Heilongjiang (8) | [0.0864–0.1136]:Guizhou, Ningxia, Xinjiang, Qinghai, Gansu (5) |
| 2018 | (0.3593–0.8018]:Shanghai (1) | (0.2174–0.3593]: Guangdong, Beijing, Zhejiang, Jiangsu, Hainan, Chongqing (6) | (0.1435–0.2174]:Tianjin, Inner Mongolia, Liaoning, Fujian, Anhui, Jiangxi (6) | (0.1007–0.1435]:Hunan, Shandong, Hubei, Sichuan, Shanxi, Henan, Yunnan (7) | [0.0678–0.1007]:Shaanxi, Hebei, Xinjiang, Heilongjiang, Jilin, Guangxi, Guizhou, Ningxia, Qinghai, Gansu (10) |

levels most of the time. However, in Inner Mongolia it increased first during 1999–2001, then decreased during the tenth five-year plan period (2001–2005), followed by increasing during the twelfth five-year plan period (2011–2015) and continued to increase during 2016–2018.

The levels in Northeast China provinces have been fluctuating for 20 years, with rising in 1999–2001, declining in the tenth and eleventh five-year plan periods (2001–2010), rising in the twelfth five-year plan period (2011–2015) and declining in 2016–2018. This can be explained by the quantitative volatility [45] and short-term stimulus effect [46] of national policies.

## Sensitivity analysis

In this section, we refer to the research of Dehdasht et al. [47] and test the sensitivity of our measurements of industrial high-quality development levels based on index weights..

Assume there are $k$ primary indices, $n$ tertiary indices, and the weight of the $r$th primary index changes from $w_{A_r}$ to $w'_{A_r}$ $(r = 1, 2, \ldots .k)$.

$$w'_{A_r} = w_{A_r} + \Delta_r$$

Therefore the weights of tertiary indices contained in the primary indices change from $w_{C_i}$ to $w'_{C_i}$ according to the equal proportion method as shown in formula below:

$$w'_{c_i} = \frac{w'_{A_n}}{w_{A_n}} \cdot w_{C_i}$$

The weights of tertiary indices contained in the other primary indices change from $w_{C_j}$ to $w'_{C_j}$ as shown in formula below:

$$w'_{c_j} = \frac{1 - w'_{A_m}}{1 - w_{A_m}} \cdot w_{C_j}$$

We test the data in 2018 only, due to the large number of years involved. The degrees of weight changes are set as 0.1, 0.15 and 0.2 for each primary index. We carried out 18 tests in which the weight of each tertiary index is calculated according to formulae above, so as to obtain the rankings of industrial high-quality development levels after changing their weights. Due to limited space, only the results of Industrial benefit ($A_1$) and Innovation ability ($A_2$) are presented in Table 5.

The results show that Shanghai has stably led the way in the high-quality industrial development, ranking first in all 18 tests. The rankings of most provinces change little with weights, indicating they are insensitive to the changes of weights. Therefore we conclude that the measurement methods used in this study can robustly evaluate the levels of high-quality development of industry.

## Interprovincial spatial correlation analysis of high-quality development of industry in China

### Global spatial correlation analysis

We use Geoda software to calculate the global Moran's I index of high-quality industrial development level of China's provinces from 1999 to 2018. The results are shown in Table 6. From the calculation results, the global Moran's I index is positive in the past 20 years. In most years, the spatial autocorrelation passes the test at the 1% confidence level. In 2008, the correlation is significant at the 5% confidence level, and in 2015, the correlation is notable at the 10% confidence level.

It can be seen that the interprovincial high-quality development of industry presents obvious positive spatial correlation, that is, interprovincial high-quality development level tends to cluster, which indicates that neighboring provinces tend to have the same level of high-quality development of industry.

### Local spatial correlation analysis

In order to explore the spatial correlation of high-quality development of industry between 30 provinces, the local Moran's I of each provinces calculated. Because of the large number of provinces involved, six key years, namely 1999, 2001, 2006, 2011, 2016 and 2018, are selected for comparative analysis. Moran scatter diagram is drawn by Geoda software, as shown in Figs 3–8.

Based on Moran scatter diagram, the clustering degree of high-quality development of industry in 30 provinces can be divided into four categories. The results are shown in Table 7, in which the contents in brackets indicate the number of provinces included: ① High-High in the first quadrant, that is, the high-quality development level of the province's industry and its neighbors'; ② The second quadrant is Low-High, that is, the level of high-quality industrial development in the province is low, and the level of neighboring provinces is high; ③ The third quadrant is Low-low, that is, the high-quality industrial development level of the province and its neighboring provinces is low; ④ The fourth quadrant is High-Low, that is, the high-quality development of industry in the province is high but its neighboring provinces' is low. Among them, the High-High and Low-Low types indicate that the level of high-quality

**Table 5. Test results of sensitivity analysis for industrial benefit (A1) and innovation ability (A2).**

| Expt. No. | Description | New Weight | | | | | | Ranking |
|---|---|---|---|---|---|---|---|---|
| | | $A_1'$ | $A_2'$ | $A_3'$ | $A_4'$ | $A_5'$ | $A_6'$ | |
| 1 | $A_1' = A_1 + 0.1$ | 0.1506 | 0.2293 | 0.1355 | 0.0849 | 0.3293 | 0.0705 | Shanghai>Guangdong>Beijing>Zhejiang>Jiangsu>Hainan>Chongqing>Tianjin>Inner Mongolia>Liaoning>Jiangxi>Fujian>Anhui>Hubei>Shandong>Sichuan>Henan>Shaanxi>Yunnan>Guizhou>Xinjiang>Hebei>Guangxi>Heilongjiang>Jilin>Ningxia>Qinghai>Gansu |
| 2 | $A_1' = A_1 + 0.15$ | 0.2006 | 0.2158 | 0.1275 | 0.0799 | 0.3100 | 0.0663 | Shanghai>Guangdong>Beijing>Zhejiang>Jiangsu>Hainan>Chongqing>Tianjin>Inner Mongolia>Jiangxi>Liaoning>Fujian>Anhui>Hainan>Hubei>Shaanxi>Shandong>Sichuan>Henan>Yunnan>Shanxi>Guizhou>Xinjiang>Hebei>Guangxi>Jilin>Heilongjiang>Ningxia>Gansu>Qinghai |
| 3 | $A_1' = A_1 + 0.2$ | 0.2506 | 0.2023 | 0.1195 | 0.0749 | 0.2906 | 0.0622 | Shanghai>Guangdong>Beijing>Zhejiang>Jiangsu>Hainan>Chongqing>Jiangxi>Inner Mongolia>Tianjin>Fujian>Liaoning>Anhui>Shaanxi>Hainan>Hubei>Sichuan>Henan>Shandong>Guizhou>Yunnan>Shanxi>Xinjiang>Hebei>Guangxi>Jilin>Heilongjiang>Ningxia>Gansu>Qinghai |
| 4 | $A_2' = A_2 + 0.1$ | 0.0438 | 0.3563 | 0.1311 | 0.0821 | 0.3186 | 0.0682 | Shanghai>Guangdong>Jiangsu>Zhejiang>Beijing>Chongqing>Hainan>Tianjin>Anhui>Jiangxi>Liaoning>Inner Mongolia>Fujian>Hainan>Hubei>Shandong>Sichuan>Yunnan>Shanxi>Henan>Xinjiang>Shaanxi>Ningxia>Hebei>Guizhou>Heilongjiang>Jilin>Guangxi>Qinghai>Gansu |
| 5 | $A_2' = A_2 + 0.15$ | 0.0404 | 0.4063 | 0.1209 | 0.0757 | 0.2939 | 0.0629 | Shanghai>Guangdong>Jiangsu>Zhejiang>Beijing>Chongqing>Hainan>Anhui>Tianjin>Jiangxi>Liaoning>Inner Mongolia>Fujian>Hainan>Hubei>Shandong>Yunnan>Sichuan>Shanxi>Henan>Xinjiang>Ningxia>Shaanxi>Hebei>Guizhou>Jilin>Heilongjiang>Guangxi>Gansu>Qinghai |
| 6 | $A_2' = A_2 + 0.2$ | 0.0370 | 0.4563 | 0.1107 | 0.0693 | 0.2691 | 0.0576 | Shanghai>Guangdong>Jiangsu>Zhejiang>Beijing>Chongqing>Anhui>Tianjin>Hainan>Jiangxi>Liaoning>Hainan>Fujian>Inner Mongolia>Hubei>Shandong>Yunnan>Sichuan>Ningxia>Henan>Shanxi>Xinjiang>Shaanxi>Hebei>Guizhou>Jilin>Heilongjiang>Guangxi>Gansu>Qinghai |

**Table 6. Moran's I of high-quality development of industry from 1999–2018.**

| Year | 1999 | 2000 | 2001 | 2002 | 2003 | 2004 | 2005 | 2006 | 2007 | 2008 |
|------|------|------|------|------|------|------|------|------|------|------|
| Global Moran's index | 0.4419 | 0.3713 | 0.3506 | 0.3331 | 0.3822 | 0.3696 | 0.4653 | 0.4581 | 0.2995 | 0.2056 |
| p-value | 0.0020 | 0.0020 | 0.0030 | 0.0030 | 0.0020 | 0.0040 | 0.0010 | 0.0010 | 0.0030 | 0.0260 |
| Year | 2009 | 2010 | 2011 | 2012 | 2013 | 2014 | 2015 | 2016 | 2017 | 2018 |
| Global Moran's index | 0.4972 | 0.1950 | 0.5150 | 0.5152 | 0.3615 | 0.3487 | 0.1601 | 0.3050 | 0.3647 | 0.2510 |
| p-value | 0.0020 | 0.0020 | 0.0020 | 0.0020 | 0.0030 | 0.0030 | 0.0600 | 0.0090 | 0.0030 | 0.0060 |

development of industry presents a strong positive correlation and spatial cluster, while the Low-High and High-Low types indicate a strong negative correlation and spatial differentiation. Generally, most provinces in China belong to High-High and Low-Low types, which indicates that their spatial clustering is significant.

From the spatial distribution characteristics of spatial clustering types, the High-High type mainly includes the eastern region, with typical serial distribution characteristics, forming three major regions, namely Beijing-Tianjin, Jiangsu-Zhejiang-Shanghai and Fujian-Guangdong-Hainan. Taking their geographical advantages, these regions strengthen the ability of opening up to the outside world, and form a "pioneer" area of high-quality development of industry by introducing abundant resources. The High-High concentration provinces changed little, but the overall scope has reduced. In 2001, the region excludes Anhui, Hubei and Hunan provinces, and in 2011, the region excludes Beijing and Guangdong Province.

The provinces with Low-Low clustering are mainly distributed in the central and western regions, and also have obvious continuous distribution characteristics. The regions

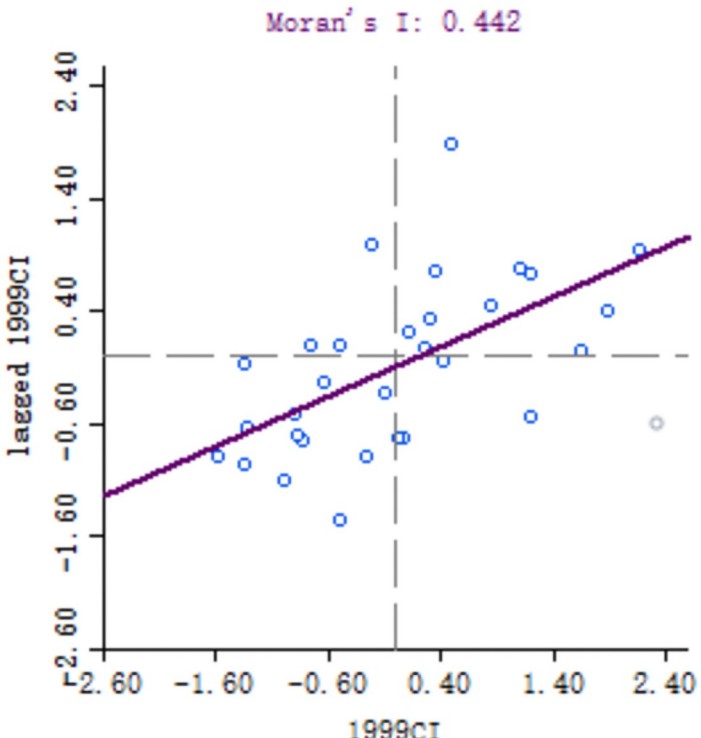

**Fig 3. Moran scatter chart of industrial high quality development level in 1999.**

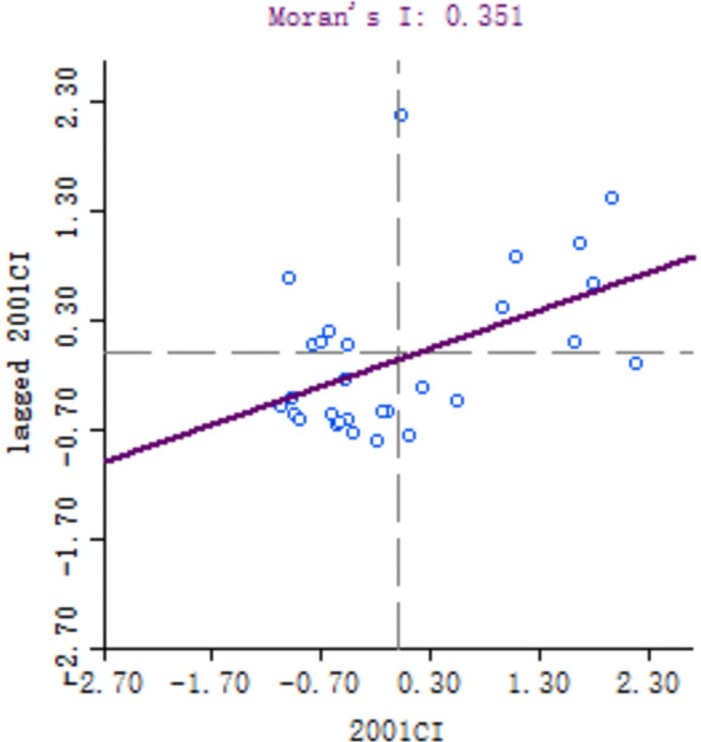

**Fig 4. Moran scatter chart of industrial high quality development level in 2001.**

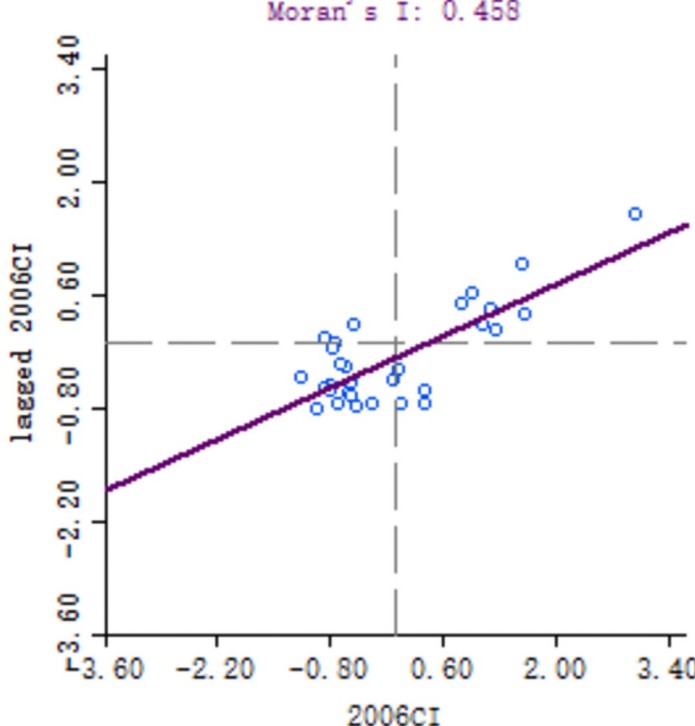

**Fig 5. Moran scatter chart of industrial high quality development level in 2006.**

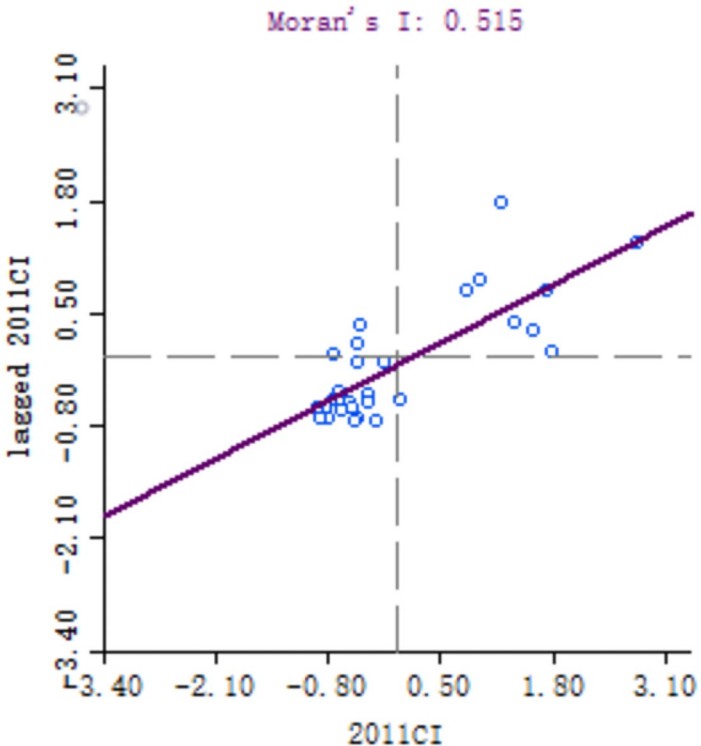

**Fig 6. Moran scatter chart of industrial high quality development level in 2011.**

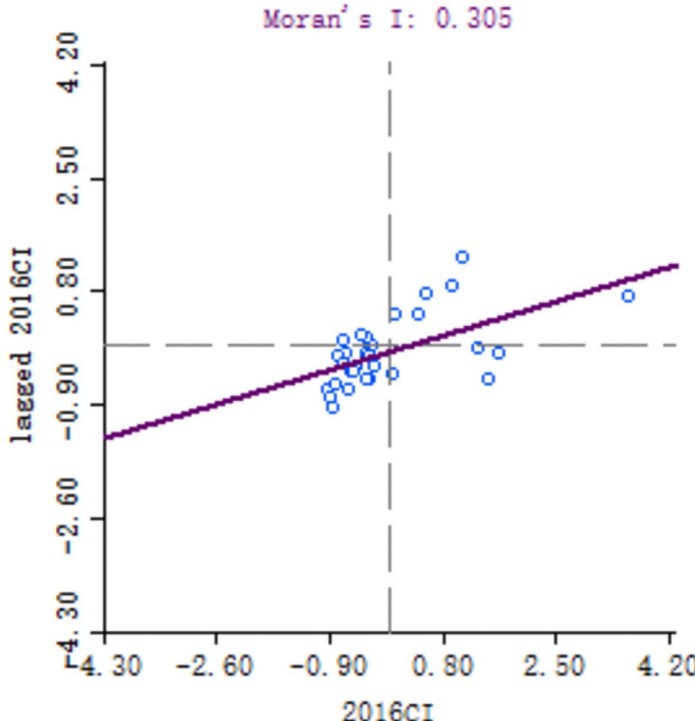

**Fig 7. Moran scatter chart of industrial high quality development level in 2016.**

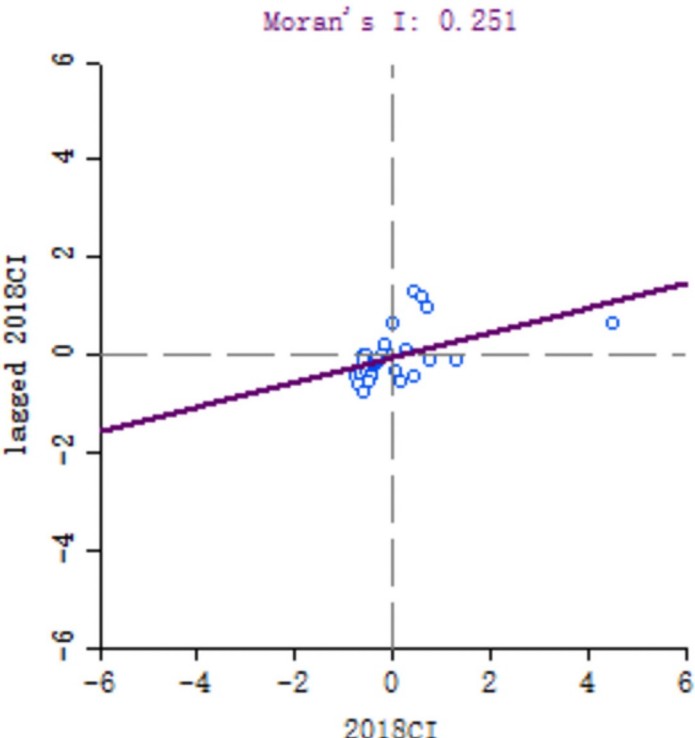

**Fig 8. Moran scatter chart of industrial high quality development level in 2018.**

represented by Shaanxi-Gansu-Ningxia-Qinghai-Xinjiang and Sichuan-Guizhou and Shanxi-Henan-Hubei are formed. The Northeast of China is included, but its not stable. Because of closed geographic condition, their development is restricted in all dimensions.

The provinces with Low-High and High-Low differences are distributed in a decentralized way. The provinces with Low-High type consist of Anhui, Guangxi, Jiangxi and Hebei. These provinces are all located around the provinces with high levels of high-quality development of industry. Therefore, they can improve their development level by using the spatial spillover effect from neighbor provinces in the future. The provinces in High-Low differences involve Beijing, Shandong, Guangdong, Chongqing and Yunnan. These regions should elaborate the driving effect and improve the high-quality development of industry in surrounding areas through industrial and resource transformation and improve the overall coordination ability.

According to the theory of Space-time Transition proposed by Rey et al. (2006), the transition of industrial cluster types in China can be divided into four categories. Type I is that the spatial clustering type of the provinces is unchanged, and the type of neighboring provinces has been transformed. This category is shown in the Moran scatter map as the vertical quadrant change, that is, the first and fourth, the second and the third quadrants change; Type II is the change of the spatial clustering type of its own but the type of neighboring provinces remains unchanged. This type is shown in Moran scatter map as the quadrant change in the horizontal, i.e. the first and second, the third and fourth quadrants change. In type III, the spatial clustering of itself and that of the neighboring provinces all have a transition, which is shown in the Moran scatter map as the oblique upward quadrant change, namely, the quadrant changes between the first and third, the second and fourth. Type IV means that the spatial clustering of itself and that of the neighboring provinces all remain unchanged and its position

**Table 7. Clustering types of high-quality industrial development in various provinces in 1999, 2001, 2006, 2011, 2016 and 2018.**

| Year | Type | Province (The number of provinces is in the bracket) |
|---|---|---|
| 1999 | High-High | Anhui, Beijing, Fujian, Guangdong, Hainan, Hubei, Hunan, Jiangsu, Shanghai, Tianjin, Zhejiang (11) |
| | Low-High | Guangxi, Hebei, Jiangxi (3) |
| | Low-Low | Gansu, Guizhou, Henan, Jilin, Inner Mongolia, Ningxia, Qinghai, Shaanxi, Shanxi, Sichuan, Xinjiang, Yunnan (12) |
| | High-Low | Chongqing, Heilongjiang, Liaoning, Shandong (4) |
| 2001 | High-High | Beijing, Fujian, Hainan, Jiangsu, Shanghai, Tianjin, Zhejiang (7) |
| | Low-High | Anhui, Guangxi, Hebei, Jiangxi, Jilin (5) |
| | Low-Low | Chongqing, Gansu, Guizhou, Henan, Hubei, Hunan, Liaoning, Inner Mongolia, Ningxia, Qinghai, Shaanxi, Shanxi, Sichuan, Xinjiang (14) |
| | High-Low | Guangdong, Heilongjiang, Shandong, Yunnan (4) |
| 2006 | High-High | Beijing, Fujian, Guangdong, Hainan, Jiangsu, Shanghai, Tianjin, Zhejiang (8) |
| | Low-High | Anhui, Guangxi, Jiangsu (3) |
| | Low-Low | Chongqing, Gansu, Guizhou, Hebei, Henan, Hubei, Hunan, Jilin, Liaoning, Inner Mongolia, Ningxia, Qinghai, Shaanxi, Shanxi, Sichuan (15) |
| | High-Low | Heilongjiang, Shandong, Xinjiang, Yunnan (4) |
| 2011 | High-High | Beijing, Fujian, Guangdong, Hainan, Jiangsu, Shanghai, Tianjin, Zhejiang (8) |
| | Low-High | Anhui, Guangxi, Hebei, Jiangxi (4) |
| | Low-Low | Gansu, Guizhou, Heilongjiang, Henan, Hubei, Hunan, Jilin, Liaoning, Inner Mongolia, Ningxia, Qinghai, Shaanxi, Shandong, Shanxi, Sichuan, Xinjiang, Yunnan (17) |
| | High-Low | Chongqing (1) |
| 2016 | High-High | Fujian, Hainan, Jiangsu, Shanghai, Tianjin, Zhejiang (6) |
| | Low-High | Hebei, Hunan, Jiangxi |
| | Low-Low | Anhui, Gansu, Guangxi, Guizhou, Heilongjiang, Henan, Hubei, Jilin, Inner Mongolia, Ningxia, Qinghai, Shaanxi, Shandong, Shanxi, Sichuan, Xinjiang, Yunnan (3) |
| | High-Low | Beijing, Chongqing, Guangdong, Liaoning (17) |
| 2018 年 | High-High | Fujian, Hainan, Jiangsu, Shanghai, Tianjin, Zhejiang (6) |
| | Low-High | Anhui, Guangxi, Hebei, Hunan, Jiangxi (5) |
| | Low-Low | Gansu, Guizhou, Heilongjiang, Henan, Hubei, Jilin, Ningxia, Qinghai, Shaanxi, Shandong, Shanxi, Sichuan, Xinjiang, Yunnan (14) |
| | High-Low | Beijing, Chongqing, Guangdong, Liaoning, Inner Mongolia (5) |

in Moran scatter map does not change. Based on this, the transition of high-quality development of industry in China can be divided into four types, as shown in Table 8.

According to the above results, more than half of China's provinces can be classified as type IV, that is, the clustering of high-quality development of industry has not undergone a

**Table 8. Types of spatial transition of high-quality industrial development level and provinces covered.**

| Type | Representative provinces |
|---|---|
| Type I: Self invariance, adjacent transition | Beijing, Inner Mongolia, Hebei, Guangdong, Guangxi, Liaoning, Anhui (2011–2016, 2016–2018), Hunan (2011–2016) |
| Type II: Self transition, adjacent invariance | Shandong, Heilongjiang, Chongqing, Yunnan, Liaoning, Anhui (1999–2001) |
| Type III: Self transition, adjacent transition | Hubei, Hunan (1999–2001) |
| Type IV: Self invariance, adjacent invariance | Tianjin, Shanghai, Jiangsu, Zhejiang, Fujian, Hainan, Shanxi, Jiangxi, Henan, Sichuan, Guizhou, Shaanxi, Gansu, Qinghai, Ningxia, Xinjiang |

transition, indicating that its spatial distribution is stable. The provinces without spatiotemporal transition locate in the eastern, central and western regions. The main spatiotemporal clustering types are High- High and Low-Low clustering types, and especially the Low-Low clustering type in the western region accounts for a large proportion. This phenomenon shows that the western region has formed a "depression" of sustained high-quality development of industry. The development of the western region in China and other regional coordination measures have not played an effective role in stimulating the development of the region. How to enhance the development energy is the key issue of these provinces.

In type I, the development level of Beijing and Guangdong is relatively high and the level of neighboring provinces fluctuates, indicating that their radiation and driving effects on their neighbors fluctuate; Hebei, Inner Mongolia, Guangxi, Liaoning, Anhui and Hunan have low development level and the level of surrounding provinces fluctuates, which indicates that the driving role of surrounding provinces to these provinces is not obvious and stable, and the regional coordinated promotion has not yet formed.

In type II, the fluctuation of Anhui Province and its surrounding areas indicate that its utilization of spillover resources needs to be improved. The industrial development in Shandong, Heilongjiang, Chongqing, Yunnan and Liaoning fluctuates while their surrounding areas are at low levels. This phenomenon shows that these provinces are still the important driving points for nearby areas and can be used in the strategy of "promote firstly and drive secondly".

## Conclusions and suggestions

### Conclusions

This paper constructs an evaluation index system of industrial high-quality development level in China from six perspectives: industrial benefit, innovation ability, coordination ability, green ability, opening ability and sharing ability. We use entropy-weight and TOPSIS to evaluate the levels of 30 provinces in China from 1999 to 2018, and lay special stress on analyzing the evaluation results in 1999, 2001, 2006, 2011, 2014 and 2018. Furthermore the robustness of this combination method is confirmed by sensitivity tests based on index weights. The main conclusions are as follows:

1. Innovation ability and open ability are the most important factors.
   Results show that, the ranking of the average weights of six primary indices in 2018 is as follows: opening ability $\succ$ innovation ability $\succ$ coordination ability $\succ$ green ability $\succ$ sharing ability $\succ$ industrial benefit. Moreover, the weights of innovation ability and opening ability have been increasing in recent years. Among the tertiary indices, number of effective invention patents per capita, proportion of high-tech industries, trade openness, openness of foreign capital, proportion of expenditure on technology introduction and employment ratio have greater weights than the others. The results indicate that innovation ability and opening ability are important indices of high-quality development, and their influence is increasing.

2. Green ability has not sufficiently contributed to China's industrial development.
   The overall change of the weights of green ability from 1999 to 2018 is insignificant. It is also ranked low in 2018. As we continue to intensify the efforts on environmental protection, energy conservation and emission reduction, their contribution to industrial high-quality development is still not enough when compared with innovation, opening, industrial benefit and other indices.

3. Regional and time evolution differences are significant.
   There are significant spatial differences in the levels of high-quality industrial development across China's provinces. While high in municipalities and coastal provinces, there is a pattern of "high in the East, medium in the middle and northeast, and low in the northwest". Their time evolutions are different as well. The levels in the southeast coastal area were continuously high, while the differences among the provinces in North China were large. In the central region the levels are found reducing first and then rising, while in the western region they increased first and then decreased, and in the northeast region they fluctuated greatly.

4. There is a significant and stable spatial clustering effect in the high-quality development of industry among China's provinces.
   Overall the high-quality development of industry among China's provinces shows positive spatial correlation. Most provinces in China are in High-High and Low-Low clustering States, which indicates that there is a significant spatial dependence effect. The High-High clustering type is mainly distributed in the eastern region and the Low-Low clustering type is mainly distributed in the western and central regions. The Low-High and High-Low differences types present divergent distribution. And the type of spatial clustering doesn't change in more than half of the provinces.

## Suggestions

Based on the above conclusions, we put forward the following four suggestions for the high-quality development of industry in China:

1. Promote opening ability and innovation ability.
   The improvement of industrial opening ability needs the joint efforts of enterprises and the government. Enterprises should reinforce the absorption of foreign capital and technology and enhance their scale and strength by absorbing capital and technology efficiently. Moreover, they should increase the export of industrial products, expand their market, and make full use of foreign market resources. The government should delegate power, and streamline and improve their services to optimize business environment and attract foreign investment. Meanwhile they need to accelerate the construction of the Belt and Road Initiative, adhere to the principle of mutual benefit and Win-win strategy, and promote the cooperation with international capacities to improve the utilization of domestic capacity.
   To promote innovation ability, enterprises should increase the investment of funds and personnel for scientific research, establish more R&D institutions, pay more attention to patented technology and improve their ability to control key technologies, so as to transform technological achievements into new product benefits. Meanwhile the government should further encourage innovation incentives, strengthen the protection of intellectual property rights and increase research investment to provide plenty of support for scientific and technological innovations.

2. Upgrade the requirement of green ability in industrial development significantly.
   Enterprises should build a low-carbon green production system with less resource consumption and less environmental pollution, reduce energy and water consumption through the introduction of advanced technology, and centralize the treatment of three wastes, so as to improve recycling efficiency to the greatest extent. Meanwhile the government should improve environmental protection law and regulation system, raise the threshold for energy conservation and environmental protection, and impose stricter punishment for the enterprises that violate the laws and regulations.

3. Promote balanced development of China's industries across regions.

   At present, there is an obvious regional imbalance in the levels of industrial high-quality development in China. In order to alleviate this problem, we should thoroughly implement the strategy of regional coordinated development, accelerate the adjustment and transformation of industrial bases in northeast China, focus on improving the innovation ability in the eastern region, implement the modernization strategy and promote the industrial cooperation between east and west regions, so as to realize an orderly industrial transfer.

4. Make full use of the spatial spillover effect to help the provinces with low-level industrial high-quality development.

   Strengthen the radiation effect of provinces with levels of industrial high-quality development and enhance the spatial spillover and diffusion effect through exchanges and win-win cooperation. First of all, focus on the provinces with high levels of development within the region to promote the whole region, such as Chongqing and Yunnan in Southwest China. Secondly, radiate the driving force Beijing and Tianjin, as the main driving points to Hebei and Inner Mongolia, and From Shanghai, Jiangsu, Zhejiang and other key provinces to the central region and southwest region.

## Supporting information

**S1 Table. Relative nearness degree and rank of 30 provinces from 1999 to 2018.**
(DOCX)

## Author Contributions

**Conceptualization:** Yuexiang Yang, Lei Ren.

**Data curation:** Lei Ren.

**Formal analysis:** Lei Ren.

**Funding acquisition:** Yuexiang Yang.

**Investigation:** Lei Ren.

**Methodology:** Yuexiang Yang, Lei Ren.

**Project administration:** Yuexiang Yang.

**Resources:** Yuexiang Yang, Lei Ren.

**Software:** Lei Ren.

**Supervision:** Yuexiang Yang, Lei Ren, Zhihui Du, Guanqun Tong.

**Validation:** Yuexiang Yang, Lei Ren, Zhihui Du, Guanqun Tong.

**Visualization:** Lei Ren.

**Writing – original draft:** Lei Ren, Zhihui Du.

**Writing – review & editing:** Yuexiang Yang, Lei Ren, Zhihui Du, Guanqun Tong.

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
