## [Decision Letter · Decision Letter 0]

11 Jun 2021

PONE-D-21-12827

Dynamic Evaluation of High-quality Development of Industry in China Based on Entropy-Weight and TOPSIS

PLOS ONE

Dear Dr. Du,

Thank you for submitting your manuscript to PLOS ONE. After careful consideration, we feel that it has merit but does not fully meet PLOS ONE’s publication criteria as it currently stands. Therefore, we invite you to submit a revised version of the manuscript that addresses the points raised during the review process.

We look forward to receiving your revised manuscript.

Kind regards,

Dragan Pamucar

Academic Editor

PLOS ONE

Journal Requirements:

We note that one or more of the authors are employed by a commercial company: Elstree Group Holdings Limited.

2.1. Please provide an amended Funding Statement declaring this commercial affiliation, as well as a statement regarding the Role of Funders in your study. If the funding organization did not play a role in the study design, data collection and analysis, decision to publish, or preparation of the manuscript and only provided financial support in the form of authors' salaries and/or research materials, please review your statements relating to the author contributions, and ensure you have specifically and accurately indicated the role(s) that these authors had in your study. You can update author roles in the Author Contributions section of the online submission form.

2.2. Please also provide an updated Competing Interests Statement declaring this commercial affiliation along with any other relevant declarations relating to employment, consultancy, patents, products in development, or marketed products, etc.  

4. We note that Figures 3-8 in your submission contain map images which may be copyrighted. All PLOS content is published under the Creative Commons Attribution License (CC BY 4.0), which means that the manuscript, images, and Supporting Information files will be freely available online, and any third party is permitted to access, download, copy, distribute, and use these materials in any way, even commercially, with proper attribution. For these reasons, we cannot publish previously copyrighted maps or satellite images created using proprietary data, such as Google software (Google Maps, Street View, and Earth). For more information, see our copyright guidelines: http://journals.plos.org/plosone/s/licenses-and-copyright.

4.1.    You may seek permission from the original copyright holder of Figures 3-8 to publish the content specifically under the CC BY 4.0 license. 

4.2.    If you are unable to obtain permission from the original copyright holder to publish these figures under the CC BY 4.0 license or if the copyright holder’s requirements are incompatible with the CC BY 4.0 license, please either i) remove the figure or ii) supply a replacement figure that complies with the CC BY 4.0 license. Please check copyright information on all replacement figures and update the figure caption with source information. If applicable, please specify in the figure caption text when a figure is similar but not identical to the original image and is therefore for illustrative purposes only.

Reviewers' comments:

Reviewer's Responses to Questions

**Comments to the Author**

1. Is the manuscript technically sound, and do the data support the conclusions?

Reviewer #1: Yes

Reviewer #2: Partly

2. Has the statistical analysis been performed appropriately and rigorously? 

Reviewer #1: Yes

Reviewer #2: No

3. Have the authors made all data underlying the findings in their manuscript fully available?

Reviewer #1: Yes

Reviewer #2: No

4. Is the manuscript presented in an intelligible fashion and written in standard English?

Reviewer #1: Yes

Reviewer #2: No

5. Review Comments to the Author

Reviewer #1: Presented methodology has a great potential in decision making process and I am giving support to the authors for investigation this topic. The strengths of this paper are: Relevant topic; Flow of the paper; and Explanation of the model. However, the author(s) need to consider the following points as limitation or further scope for refining the paper:

- Rewrite Abstract - Abstract should cover five main elements, introduction, problem statement, methodology, contributions and results.

- Introduction should be clearly stated research questions and targets first. Why is the topic important (or why do you study on it)? What are your contributions? Why is to propose extended TOPSIS methodology. You should provide more information in this regard and in more coherent way.

- Need to highlight the novelty of study in the introduction and motivation.

- Literature review (LR) should be presented in a better way.

- You should provide flowchart and organize the section according the stages presented on flowchart.

- What about validation and comparisons with existing approaches. What are advantages of this approach? What are its limitations?

Reviewer #2: The article has many fundamental problems. The following are some of my comments in order to improve the technicality and readability of the paper:

1-The manuscript neither presents the initial data collected nor shows the process of data analysis.

2-The manuscript does not check the reliability and validity of the achieved results (for example, comparison against existing alternative methods).

3-The combination of the two methods entropy and TOPSIS is not sufficiently exposed; the use of these methods to the application domain must be better explicated.

4- The analysis of the results is really poor. The conclusions are trivial and it is difficult to understand what the add-value of this approach is.

5-The discussion part is only an explanation of the result and not a discussion (without a real analysis of them).

6-The submission is badly written and the authors need to get an independent native speaker to proof-read the entire manuscript.

Overall, the paper has some potential; however, the current form does not offer any novelty and robustness.

6. PLOS authors have the option to publish the peer review history of their article (what does this mean?). If published, this will include your full peer review and any attached files.

Reviewer #1: No

Reviewer #2: **Yes: **Dr. Gholamreza Dehdasht

---

## [Author Response · Author response to Decision Letter 0]

14 Jul 2021

A Reply for Comments and Suggestions of the Manuscript ID (PONE-D-21-12827)

Dear Editors and reviewers, 

We first thank the reviewers and editors sincerely for giving very helpful suggestions and comments to improve our manuscript. Our revisions have been made according to the questions and suggestions from two reviewers. The detailed revisions are as follows:

Questions and suggestions from Reviewer 1: 

Presented methodology has a great potential in decision making process and I am giving support to the authors for investigation this topic. The strengths of this paper are: Relevant topic; Flow of the paper; and Explanation of the model. However, the author(s) need to consider the following points as limitation or further scope for refining the paper:

[QS1] Rewrite Abstract - Abstract should cover five main elements, introduction, problem statement, methodology, contributions and results. 

[A1] We have rewritten the abstract as the reviewer suggested. The specific amendments are as follows. 

China’s economy has been transitioning from a phase of rapid growth to high-quality development. The high-quality development of industry is the foundation of a sustainable and healthy growth of national economy, and is of great significance to improve people’s living standards, and to meet people's needs for a better life. We develop an evaluation index system of high-quality development of industry from the perspectives of industrial benefit, innovation ability, coordination ability, green ability, opening ability and sharing ability. Based on a panel data of 30 provinces in China during 1999-2018, we evaluate the level of high-quality development of industry using the entropy-weight method and Technique for Order Preference by Similarity to an Ideal Solution (TOPSIS) method. Meanwhile we select six specific years and adopt the Natural Breaks method to classify the provinces according to their levels. At last, Moran's I index is used to analyze the spatial correlation among the provinces. Our results show that opening ability and innovation ability are found to have greater impacts on industrial high-quality development than other indices, and their influence has been increasing in recent years. There are large spatial and temporal differences among different provinces. Municipalities and coastal provinces are found to be at constantly high levels. The levels in the central region dropped first and then increased, however it was the opposite in the western region. In the northeast region, the levels fluctuated greatly. Overall the high-quality development of industry among China's provinces shows positive spatial correlation. Most provinces in China are in High-High and Low-Low clustering States. The High-High clustering type is mainly distributed in the eastern region and the Low-Low clustering type is mainly distributed in the western and central regions. Our conclusions are: (1) Innovation ability and open ability are the most important factors. (2) Green ability has not sufficiently contributed to China's industrial development. (3) Regional and time evolution differences are significant. (4) There is a significant and stable spatial clustering effect in the high-quality development of industry among China's provinces. .

[QS2] Introduction should be clearly stated research questions and targets first. Why is the topic important (or why do you study on it)? What are your contributions? Why is to propose extended TOPSIS methodology. You should provide more information in this regard and in more coherent way.

[A2] We condensed the introduction in more coherent way and added some relative materials to improve the clarity of our research questions and contributions.

The importance of this topic: In 2017, the 19th National Congress of the Communist Party of China proposed the concept of high-quality development for the first time, indicating that China’s economy has been transitioning from a phase of rapid growth to high-quality development. “High-quality development is the development which can well meet people's growing needs for a better life. It is the development that embodies a new concept that innovation becomes the first driving force, coordination becomes the endogenous characteristic, green becomes the universal form, opening becomes the inevitable course, and sharing becomes the fundamental purpose.” General secretary of the CPC Central Committee Xi Jinping said. From this, we can see that high-quality development means strong innovation ability, green environmental protection, harmonious social and economic relationship, and sharing of development achievements by the people, on top of the traditional concept of high input-output efficiency. General Secretary Xi Jinping also stressed that the requirements of high-quality development should be reflected in the economic, social, cultural, ecological and all other fields. The high-quality development of industry is the foundation of a sustainable and healthy growth of national economy, and is of great significance to improve people’s living standards, and to meet people's needs for a better life.

The contribution of this study: This paper originates in constructing a comprehensive index system and evaluating the high-quality development of industry from provincial and time dimensions, and reveal the interprovincial correlation among the provinces. We developed important references to promoting the high-quality development of industry in China.

Why do we propose Entropy weight TOPSIS? Comprehensive evaluation methods include subjective weighting and objective weighting methods. The objective weighting methods determine the weights based on actual data and have the advantages of objectivity and transparency. They are applicable to our study since it is based on a large number of empirical data,. As one of the most widely used objective weighting methods, entropy weight method determines the weight of all data information, which is more complete than the commonly used Principal Component Analysis [1]. Moreover the entropy weight method does not require too much data, avoiding the shortcomings of the commonly used factor analysis which requires large quantities of sample data. In addition, we introduce the TOPSIS to rank the high-quality development of industry of each province. Some researches show that the Entropy-weight TOPSIS method has better stability and robustness than the other methods. According to the analysis conducted by Zhang [61], when using the mean value of standard deviation as a measure of stability when the number of evaluation objects, indices or both change, Entropy-weight method is the best evaluation method for stability in the research area and TOPSIS method has good performance in the ranking method. Zhang Lijun [62] use Kendall harmony coefficient to analyze the robustness of different comprehensive evaluation methods when the dimensionless method and sample size changed. The results testify the high robustness of Entropy-weight method.

[QS3] Need to highlight the novelty of study in the introduction and motivation.

[A3] We highlight the novelty of the study by modifying the introduction. It mainly includes: enriching and polishing the limitations, main contents and contributions of the research on the measurement of high-quality development of industry introduced in the introduction.

There are some limitations in the measurement of high-quality development of industry. First, the Five Concepts of Development are the most consistent with China’s high-quality development, which provide a reference for the construction of our comprehensive index system. The second is that the entropy weight and its improved methods are most widely used in the evaluation of levels of indices. Third, current research is lack of analysis from time and space dimensions, which cannot grasp the dynamic trend and spatial characteristics of high-quality development of industry in China. Fourth, current research is mainly based on a specific region or the whole country, and pays no attention to interprovincial analysis

Our study mainly bases on the Five Concepts for Development and builds evaluation index system for high-quality development of industry by introducing industrial performance indicators system. We measure the high-quality development of industry 30 provinces from 1999 to 2018 by the entropy weight and TOPSIS method and we use the spatial correlation analysis to study its spatial characteristics, so as to reveal the spatial distribution and temporal evolution regularity, and then we put forward targeted countermeasures and suggestions. This paper makes up for the lack of current research to evaluate the high-quality development of industry from the provincial and time dimensions, and has important reference value for promoting the high-quality development of industry in China.

[QS4] Literature review (LR) should be presented in a better way.

[A4] This paper mainly includes the literature review of the overall research on high-quality development, which focuses on the literature review of high-quality development of economy and logically sorts the research status of high-quality development of industry, enriching and refining its limitations. We added the survey of the research status of Entropy weight TOPSIS used in this study.

[QS5] You should provide flowchart and organize the section according the stages presented on flowchart.

[A5] We introduce the organization of the article in the last paragraph of the introduction. The article is organized as follows. Section 1 reviews the current research on the measurement of high-quality development. Section 2 constructs the evaluation index system for high-quality development of industry. Section 3 introduces the implementation steps of entropy weight TOPSIS method and the calculation method of spatial correlation analysis. Empirical results are reported in Section 4, including the introduction of data sources and methods, the evaluation results of industrial high-quality development level and the results of spatial correlation analysis. Section 5 summarizes the conclusion, and puts forward targeted policy suggestions.

[QS6] What about validation and comparisons with existing approaches. What are advantages of this approach? What are its limitations?

[A6] In introduction, we introduced the characteristics of Entropy weight TOPSIS and its research status in high-quality development evaluation and other fields. In Section 3, we focused on the applicability of this method to the problems and the comparison between Entropy weight TOPSIS and other measurement methods in measuring objects, stability and robustness.

We consider using other comprehensive evaluation methods to evaluate the high-quality development of industry in China, and compare others with the Entropy weight TOPSIS method used in this paper, objective analysis methods such as principal component analysis, factor analysis, or subjective analysis methods such as analytic hierarchy process. The best evaluation method can be selected by comparing the robustness of different methods like the best scheme consistency rate, scheme ranking matching rate, mean square error, mean absolute difference, Spearman coefficient, Kendall coefficient and Kendall harmony coefficient.

However, the focus of the article will be papered over if include the results of comparison. Adding more comprehensive evaluation method comparison will lead to the focus of the paper from the analysis of high-quality development status to the discussion of comprehensive evaluation method. In addition, the comparison of various methods will lead to a long paper.

Therefore, we decided to refer to the current academic research results of comprehensive evaluation methods to help the comparative analysis of evaluation methods.

 First of all, we analyze the research status of the measurement of high-quality economic development and high-quality industrial development, and draw a conclusion that although the measurement methods have not been unified, the entropy weight method and its improved methods are the most widely used objective evaluation methods. Secondly, we compare the subjective weighting method with the objective weighting method to explain the reason why we choose the objective evaluation method. Secondly, we compare with other commonly used objective evaluation methods to explain the advantages and applicability of entropy weight TOPSIS method. Finally, we analyze the robustness and stability of the comprehensive evaluation method based on the existing research results.

Comprehensive evaluation methods include subjective weighting and objective weighting methods. The objective weighting methods determine the weights based on actual data and have the advantages of objectivity and transparency. They are applicable to our study since it is based on a large number of empirical data. As one of the most widely used objective weighting methods, entropy weight method determines the weight of all data information, which is more complete than the commonly used Principal Component Analysis [1]. Moreover, the entropy weight method does not require too much data, avoiding the shortcomings of the commonly used factor analysis which requires large quantities of sample data [2]. In addition, we introduce the TOPSIS to rank the high-quality development of industry of each province. Some researches show that the Entropy-weight TOPSIS method has better stability and robustness than the other methods. According to the analysis conducted by Zhang [61], when using the mean value of standard deviation as a measure of stability when the number of evaluation objects, indices or both change, Entropy-weight method is the best evaluation method for stability in the research area and TOPSIS method has good performance in the ranking method. Zhang Lijun [62] use Kendall harmony coefficient to analyze the robustness of different comprehensive evaluation methods when the dimensionless method and sample size changed. The results testify the high robustness of Entropy-weight method.

Referring to the characteristics of the research problem and combined with the applicability and stability of each method, we use Entropy weight method to weight the indicators, and combine it with TOPSIS method to rank the high-quality development of industry in each province. In addition, based on the comprehensive evaluation of high-quality industrial development level, this paper also studies its spatial correlation, so as to obtain the spatial characteristics of high-quality development of industry in China.

Questions and Suggestions from Reviewer 2

The article has many fundamental problems. The following are some of my comments in order to improve the technicality and readability of the paper:

[QS1] The manuscript neither presents the initial data collected nor shows the process of data analysis.

[A1] We will upload the original data and calculation process data of the paper.

Due to the limitation of space and the huge amount of data, it is difficult to show the data analysis process in detail in the paper. In our paper, we made a detailed explanation of the specific calculation steps of Entropy weight TOPSIS, and showed the calculation results of our Entropy weight method and the progress and ranking of high-quality development of industry in six key years calculated by TOPSIS. Meanwhile, detailed results of relative nearness degrees and rankings of 30 provinces from 1999 to 2018 are presented in supplementary material.

[QS2] The manuscript does not check the reliability and validity of the achieved results (for example, comparison against existing alternative methods).

[A2] In introduction, we introduced the characteristics of Entropy weight TOPSIS and its research status in high-quality development evaluation and other fields. In Section 3, we focused on the applicability of this method to the problems and the comparison between Entropy weight TOPSIS and other measurement methods in measuring objects, stability and robustness.

We consider using other comprehensive evaluation methods to evaluate the high-quality development of industry in China, and compare others with the Entropy weight TOPSIS method used in this paper, objective analysis methods such as principal component analysis, factor analysis, or subjective analysis methods such as analytic hierarchy process. The best evaluation method can be selected by comparing the robustness of different methods like the best scheme consistency rate, scheme ranking matching rate, mean square error, mean absolute difference, Spearman coefficient, Kendall coefficient and Kendall harmony coefficient.

However, the focus of the article will be papered over if using this method of comparison. Adding more comprehensive evaluation method comparison will lead the focus of the paper from the analysis of high-quality development status to the discussion of comprehensive evaluation methods. In addition, the comparison of various methods will lead to a lengthy paper.

Therefore, we decided to refer to the current academic research results of comprehensive evaluation methods to help the comparative analysis of evaluation methods.

 First of all, we analyze the research status of the measurement of high-quality economic development and high-quality industrial development, and draw a conclusion that although the measurement methods have not been unified, the entropy weight method and its improved methods are the most widely used objective evaluation methods. Secondly, we compare the subjective weighting method with the objective weighting method to explain the reason why we choose the objective evaluation method. Secondly, we compare with other commonly used objective evaluation methods to explain the advantages and applicability of entropy weight TOPSIS method. Finally, we analyze the robustness and stability of the comprehensive evaluation method based on the existing research results.

In terms of comprehensive evaluation, the current commonly used methods are subjective weighting method and objective weighting method. The objective weighting method is based on the actual data to determine the weight, which has the advantage of objectivity and transparency. Based on a large number of empirical data, the objective weighting method has broader applicability.

Entropy weight method, as the most widely used high-quality development evaluation method, has broader applicability for this research problem. Firstly, Entropy weight method is used to determine the weight of all data information. Compared with the commonly used principal component analysis method, the information is more complete, and it is more suitable for the evaluation high-quality development of industry in multiple years. Secondly, the Entropy weight method does not require too much data avoiding the shortcomings of the commonly used factor analysis method. In addition, this paper also introduces the TOPSIS to rank the high-quality development of industry among provinces. This method is simple in calculation and reasonable in results and improves the accuracy of high-quality development of industry evaluation.

In addition, the research shows that the Entropy weight TOPSIS method has better stability and robustness than other methods. According to the analysis of the stability comparison of comprehensive evaluation methods by Zhang [3], the mean value of the standard deviation of the evaluation when the number of evaluation objects changes, the number of indexes changes or both changes is used as the measurement index of the stability. The results show that the Entropy weight method is the best evaluation method for the stability in the research scope, and the TOPSIS also has a good performance in the rank method. Zhang uses Kendall harmony coefficient to analyze the robustness of different comprehensive evaluation methods when the sample size changes and the dimensionless method. The results show that the Entropy weight has high robustness.

Referring to the characteristics of the research problem and combined with the applicability and stability of each method, we use Entropy weight method to weight the indicators, and combine it with TOPSIS method to rank the high-quality development of industry in each province. In addition, based on the comprehensive evaluation of high-quality industrial development level, this paper also studies its spatial correlation, so as to obtain the spatial characteristics of high-quality development of industry in China.

[QS3] The combination of the two methods Entropy and TOPSIS is not sufficiently exposed; the use of these methods to the application domain must be better explicated.

[A3] In introduction, we introduced the characteristics of Entropy weight TOPSIS and its research status in high-quality development evaluation and other fields. In Section 3, we focused on the applicability of this method to the problems and the comparison between Entropy weight TOPSIS and other measurement methods in measuring objects, stability and robustness.

Specifically, the objective weighting method is more applicable and transparent for a large number of empirical data. Entropy weight method, as the most widely used high-quality development evaluation method, has broader applicability for this research problem. Firstly, Entropy weight method is used to determine the weight of all data information. Compared with the commonly used principal component analysis method, the information is more complete, and it is more suitable for the evaluation high-quality development of industry in multiple years. Secondly, the Entropy weight method does not require too much data avoiding the shortcomings of the commonly used factor analysis method. In addition, this paper also introduces the TOPSIS to rank the high-quality development of industry among provinces. This method is simple in calculation and reasonable in results and improves the accuracy of high-quality development of industry evaluation.

We introduced the TOPSIS to rank the high-quality development of industry among provinces. This method is simple in calculation and reasonable in results and can improve the accuracy of high-quality development of industry evaluation. The core idea of entropy weight TOPSIS is that on the basis of standardizing each measurement index, entropy weight method is used to give each measurement index weight value, and then TOPSIS method is used to quantify and sort the high-quality industrial development level of each province, so as to realize the combination of the advantages of Entropy weight and TOPSIS method.

Because of the advantages illustrated above, this method has been widely used in academia. For example, Wei and Li, Du et al., Ou et al., Wang et al. used it in the field of high-quality development evaluation. In addition, it is also widely used in sustainable development evaluation, efficiency, contribution, people's quality of life, competitiveness evaluation, decision-making research and other aspects.

[QS4] The analysis of the results is really poor. The conclusions are trivial and it is difficult to understand what the add-value of this approach is.

[A4] Because there are many provinces and years involved, we optimized the method through a series of measures. In the study, we first analyze the indices in our evaluation system, as to find which indices play greater roles in the evaluation of industrial high-quality development. Secondly, six key years are selected for in-depth study. We used the Natural Breaks to divide their development level into five levels. We divide provinces according to regions for analysis. It mainly includes the analysis of the regional distribution pattern of high-quality industry development and the evolution of the high-quality development of each region. Through the selection of key years and the division by regions, we formed the logic of analysis and presented our discussion. To enrich our contribution we added the spatial correlation analysis in this manuscript.

[QS5] The discussion part is only an explanation of the result and not a discussion (without a real analysis of them).

[A5] This paper involves many provinces and years. The index system constructed in this study includes six dimensions of "economic benefit, innovation, coordination, green, opening and sharing". In order to make a deeper analysis of the high-quality development level of industry, we added the spatial correlation analysis in this revision, and enhanced the research depth by increasing the analysis dimensions.

[QS6] The submission is badly written and the authors need to get an independent native speaker to proof-read the entire manuscript.

[A6] We carefully polished and proofread the manuscript.

We promise that all authors agree to the final version, and the authors' names and affiliations are correct, and that funding sources are correctly acknowledged.

Thank you again for your suggestions. Please feel free to contact us if you have any questions.

Best regards,

Yuexiang Yang, Lei Ren, Zhihui Du, Guanqun Tong

---

## [Decision Letter · Decision Letter 1]

23 Aug 2021

PONE-D-21-12827R1

Measurement and Spatiotemporal Analysis of High-quality Development of China's Industry

PLOS ONE

Dear Dr. Du,

Thank you for submitting your manuscript to PLOS ONE. After careful consideration, we feel that it has merit but does not fully meet PLOS ONE’s publication criteria as it currently stands. Therefore, we invite you to submit a revised version of the manuscript that addresses the points raised during the review process.

We look forward to receiving your revised manuscript.

Kind regards,

Dragan Pamucar

Academic Editor

PLOS ONE

Journal Requirements:

Reviewers' comments:

Reviewer's Responses to Questions

**Comments to the Author**

1. If the authors have adequately addressed your comments raised in a previous round of review and you feel that this manuscript is now acceptable for publication, you may indicate that here to bypass the “Comments to the Author” section, enter your conflict of interest statement in the “Confidential to Editor” section, and submit your "Accept" recommendation.

Reviewer #1: All comments have been addressed

Reviewer #2: (No Response)

2. Is the manuscript technically sound, and do the data support the conclusions?

Reviewer #1: Yes

Reviewer #2: (No Response)

3. Has the statistical analysis been performed appropriately and rigorously? 

Reviewer #1: Yes

Reviewer #2: (No Response)

4. Have the authors made all data underlying the findings in their manuscript fully available?

Reviewer #1: Yes

Reviewer #2: (No Response)

5. Is the manuscript presented in an intelligible fashion and written in standard English?

Reviewer #1: Yes

Reviewer #2: (No Response)

6. Review Comments to the Author

Reviewer #1: The authors have addressed the point of my concern. I am happy with their corrections. Hence, I would like to recommend this manuscript to be published.

Reviewer #2: 1. Despite the authors' explanations, the reviewer is not yet convinced that the the reliability and validity of the proposed hybrid method has been well investigated. Please check the similar articles that have been published in PlosOne

https://doi.org/10.1371/journal.pone.0228746

https://doi.org/10.1371/journal.pone.0177578

https://doi.org/10.1371/journal.pone.0201604

2. The conclusion is just listing experimental results. You should provide a clear scientific justification for your work in this section.

7. PLOS authors have the option to publish the peer review history of their article (what does this mean?). If published, this will include your full peer review and any attached files.

Reviewer #1: No

Reviewer #2: **Yes: **Gholamreza Dehdasht

---

## [Author Response · Author response to Decision Letter 1]

22 Sep 2021

A Reply for Comments and Suggestions of the Manuscript ID (PONE-D-21-12827)

Dear Editors and reviewers, 

We thank the reviewers and editors again for giving very helpful suggestions and comments to improve our manuscript. Our revisions have been made according to the questions and suggestions from two reviewers. The detailed revisions are as follows:

Questions and suggestions from Reviewer 1: 

The authors have addressed the point of my concern. I am happy with their corrections. Hence, I would like to recommend this manuscript to be published.

[R] Thanks for your help and approval.

Questions and Suggestions from Reviewer 2

[QS1] Despite the authors' explanations, the reviewer is not yet convinced that the reliability and validity of the proposed hybrid method has been well investigated. Please check the similar articles that have been published in Plos One

https://doi.org/10.1371/journal.pone.0228746

https://doi.org/10.1371/journal.pone.0177578

https://doi.org/10.1371/journal.pone.0201604

[A1] We are grateful for the references provided by reviewer. We mainly learned from the research of Dehdasht et al. and tested the robustness of the combination method proposed in this paper through the sensitivity analysis based on index weight. Because there are too many indicators involved in this study, we first changed the weight of the primary indices, then we reflect the results into the weight change of the tertiary indices with equal proportion, and finally we calculated the weight of the tertiary indicators contained in other primary indicators. For the six primary indices, we set up three degree change of weigh and conducted 18 experiments. The experimental results show the robustness of the method used in this study which can evaluate the high-quality development of industry reliably.

[QS2] The conclusion is just listing experimental results. You should provide a clear scientific justification for your work in this section.

[A2] We conducted sensitivity analysis to justify the robustness and then rationality of results.

We promise that all authors agree to the final version, and the authors' names and affiliations are correct, and that funding sources are correctly acknowledged.

Thank you again for your suggestions. Please feel free to contact us if you have any questions.

Best regards,

Yuexiang Yang, Lei Ren, Zhihui Du, Guanqun Tong

---

## [Decision Letter · Decision Letter 2]

28 Oct 2021

Measurement and Spatiotemporal Analysis of High-quality Development of China's Industry

PONE-D-21-12827R2

Dear Dr. Du,

We’re pleased to inform you that your manuscript has been judged scientifically suitable for publication and will be formally accepted for publication once it meets all outstanding technical requirements.

Kind regards,

Dragan Pamucar

Academic Editor

PLOS ONE

Additional Editor Comments (optional):

Reviewers' comments:

Reviewer's Responses to Questions

**Comments to the Author**

1. If the authors have adequately addressed your comments raised in a previous round of review and you feel that this manuscript is now acceptable for publication, you may indicate that here to bypass the “Comments to the Author” section, enter your conflict of interest statement in the “Confidential to Editor” section, and submit your "Accept" recommendation.

Reviewer #2: All comments have been addressed

2. Is the manuscript technically sound, and do the data support the conclusions?

Reviewer #2: Yes

3. Has the statistical analysis been performed appropriately and rigorously? 

Reviewer #2: Yes

4. Have the authors made all data underlying the findings in their manuscript fully available?

Reviewer #2: Yes

5. Is the manuscript presented in an intelligible fashion and written in standard English?

Reviewer #2: Yes

6. Review Comments to the Author

Reviewer #2: (No Response)

7. PLOS authors have the option to publish the peer review history of their article (what does this mean?). If published, this will include your full peer review and any attached files.

Reviewer #2: No

---

## [Editor Report · Acceptance letter]

10 Dec 2021

PONE-D-21-12827R2 

Measurement and Spatiotemporal Analysis of High-quality Development of China's Industry 

Dear Dr. Du:

I'm pleased to inform you that your manuscript has been deemed suitable for publication in PLOS ONE. Congratulations! Your manuscript is now with our production department. 

Kind regards, 

on behalf of

Dr. Dragan Pamucar 

Academic Editor

PLOS ONE